# Advances in Novel Diagnostic Techniques for Alveolar Echinococcosis

**DOI:** 10.3390/diagnostics15050585

**Published:** 2025-02-27

**Authors:** Huanhuan Liu, Yijia Xie, Xiaoyu An, Dazhuang Xu, Shundong Cai, Chengchao Chu, Gang Liu

**Affiliations:** 1State Key Laboratory of Vaccines for Infectious Diseases, Center for Molecular Imaging and Translational Medicine, Xiang An Biomedicine Laboratory, National Innovation Platform for Industry-Education Integration in Vaccine Research, Fujian Engineering Research Center of Molecular Theranostic Technology, School of Public Health, Xiamen University, Xiamen 361102, China; 2Department of Nuclear Medicine, School of Public Health, The Second Affiliated Hospital of Chengdu Medical College, China National Nuclear Corporation 416 Hospital, Chengdu 610051, China; 3Xiamen University Affiliated Xiamen Eye Center, Eye Institute of Xiamen University, Fujian Provincial Key Laboratory of Ophthalmology and Visual Science, School of Medicine, Xiamen University, Xiamen 361102, China

**Keywords:** *Echinococcus multilocularis*, alveolar echinococcosis, parasite, diagnostic techniques

## Abstract

Alveolar echinococcosis (AE), caused by the larval stage of the tapeworm *Echinococcus multilocularis*, is a serious parasitic disease that presents significant health risks and challenges for both patients and healthcare systems. Accurate and timely diagnosis is essential for effective management and improved patient outcomes. This review summarizes the latest diagnostic methods for AE, focusing on serological tests and imaging techniques such as ultrasonography (US), computed tomography (CT), magnetic resonance imaging (MRI), and positron emission tomography/computed tomography (PET/CT). Each imaging modality has its strengths and limitations in detecting and characterizing AE lesions, such as their location, size, and invasiveness. US is often the first-line method due to its non-invasiveness and cost-effectiveness, but it may have limitations in assessing complex lesions. CT provides detailed anatomical information and is particularly useful for assessing bone involvement and calcification. MRI, with its excellent soft tissue contrast, is superior for delineating the extent of AE lesions and their relationship to adjacent structures. PET/CT combines functional and morphological imaging to provide insights into the metabolic activity of lesions, which is valuable for monitoring treatment response and detecting recurrence. Overall, this review emphasizes the importance of a multifaceted diagnostic approach that combines serological and imaging techniques for accurate and early AE diagnosis, which is crucial for effective management and improved patient outcomes.

## 1. Introduction

Echinococcosis is a zoonotic parasitic disease that manifests in two main forms in humans: cystic echinococcosis (CE) and alveolar echinococcosis (AE), caused by the tapeworms *Echinococcus granulosus* and *Echinococcus multilocularis*, respectively [1,2]. Both are serious diseases, with AE being particularly severe, as it has higher disability rates and mortality compared to CE [3,4]. Additionally, due to its pathological features bearing a striking resemblance to malignant tumors, AE is often referred to as “worm cancer” [5]. AE typically occurs later than CE, with a latency period of 5–15 years [6,7]. In the later stages of hydatid disease, the mortality rate is high (>90%) if left untreated or improperly managed [8]. Only early diagnosis based on diagnostic imaging and serological markers can improve treatment outcomes [9,10].

### 1.1. Epidemiology of E. multilocularis

AE primarily occurs in the northern hemisphere, especially in China, North America, and Central Europe (southern Germany, Switzerland, Austria, and eastern France) [11,12]. AE is mainly distributed in high mountain regions and pastoral areas with developed animal husbandry [13,14]. The life cycle of *E. multilocularis* is illustrated in Figure 1. Foxes are the primary host of *E. multilocularis*, and other hosts include canines such as dogs and raccoon dogs [15,16]. The mature tapeworms parasitize the small intestines of definitive hosts, excreting eggs through feces, while the metacestode develops in the liver and other viscera of intermediate hosts [17]. Humans are accidental intermediate hosts in its life cycle, becoming infected by ingesting food or water contaminated with parasite eggs, or through close contact with infected animals [18,19]. After ingestion, the eggs hatch and release oncospheres within the intestines, which then penetrate the intestinal wall into the portal veins and lymphatic vessels, eventually reaching the liver [20]. In the liver, *E. multilocularis* larvae develop into tumor-like buds that evolve into multiple vesicles containing a germinal layer surrounded by layered membranes [21]. A vigorous granulomatous reaction produced by the host’s immune system surrounds the lesion. Primary liver involvement is the most common in AE, accounting for 98% of cases. Additionally, there are reports of metastasis to the lungs [22,23], brain [24,25,26], bones [27], and other sites in some cases [28]. It is estimated that more than 18,000 cases of AE occur annually, with 91% of these cases reported in China [29].

### 1.2. The Structure of an AE Cyst

*E. multilocularis* produces multilocular alveolar cysts ranging from 1 to 30 mm in size, with each vesicle containing a semi-solid, protein-rich matrix [30]. Its boundary is ill-defined and consists of a thin, acellular laminated membrane and an inner nucleated germinal layer [31,32]. Budding originates from the peripheral germinal layer, causing the lesion’s edge to infiltrate surrounding parenchymal and extraparenchymal tissues [33]. The lesions may be solitary or multiple, and can be large (up to 15–20 cm). The aggregated vesicles (small cysts less than 1 cm) give the lesion a multilocular appearance. Many vesicles cluster together and grow like tumors, infiltrating the surrounding parenchyma, leading to chronic inflammation in the host. This results in the cyst being surrounded by abundant immune cells, macrophages, and new blood vessels. Due to the chronic inflammation, fibrous tissue and calcification are observed around the cysts [34]. There is no clear delineation between the parasitic tissue and the adjacent normal hepatic parenchyma, and the hepatic parenchyma near the mass is often atrophic, with retraction of the capsule due to biliary or vascular infiltration. As the size of the AE lesion increases, necrosis often occurs in the central area due to poor angiogenesis, which may result in bacterial infections and the development of central abscesses. Larval vesicles can also be released into blood vessels or lymphatic vessels, often leading to metastasis to distant organs [35].

## 2. Diagnosis

### 2.1. Serology

Alveolar echinococcosis is a serious disease, and early diagnosis is crucial for its management [36,37,38]. Underdiagnosis is suspected in many resource-limited settings worldwide [39,40]. Currently, the diagnosis of echinococcosis mainly relies on imaging, with serological tests serving as a valuable supplementary tool [41,42]. The Em2 antigen is a mucin-type glycoprotein found in the laminated layer of *E. multilocularis* and is the most specific antigen for detecting AE [43,44]. It has been established in enzyme-linked immunosorbent assays (ELISA) rather than Western blotting, because it is a naturally occurring carbohydrate rather than a protein. The immunodominant epitope of Em2 is a trisaccharide (Galα1-4Galβ1-3GalNAc), which is broadly distributed across several trematodes and demonstrates cross-reactivity [43,45].

The Em18 antigen is isolated from the metacestode of the parasite and has been proven to have high sensitivity and specificity to AE, and it has been applied in ELISA and immunoblotting [46,47]. Em18 protein is one of the most promising antigens for the serological diagnosis of AE in human patients [48,49]. However, these proteins are components or proteolytic fragments of ezrin-like proteins, possessing the same epitope from a single ELP molecule, and may cross-react with the serum of CE and other trematode diseases [50]. The use of recombinant protein Em18 (rEm18) improves the differential diagnostic performance compared to the native protein, as the correct identification of native Em18 in Western blotting may be hindered by other antigen components of similar molecular size [51]. Currently, at least one Western blot kit (Echinococcus Western Blot IgG; LDBIO Diagnostics, Lyon, France) is commercially available for routine serological diagnosis and differentiation of Echinococcus spp. [52,53,54]. This test is based on the detection of specific IgG directed against *E. multilocularis* whole larval antigen (including Em16 and Em18 antigens). Sako and colleagues have verified that Em18 is a product of cysteine metabolism of the 65-kilodalton (KDa) multilocular *E. multilocularis* surface protein, known as Em10 or Em II-3 [55]. They successfully expressed the rEm18 using the *Escherichia coli* system and demonstrated a detection rate of 90.3% in pathologically confirmed samples from AE patients. The rEm18 antigen has also been developed for use in a rapid immunochromatographic test (ICT) kit, demonstrating good sensitivity and specificity [56,57,58,59]. This kit is commercially available now (ADAMU-AE kit, ICST Co. Ltd., Saitama, Japan). At present, the commonly used ELISA for detecting AE in the laboratory involves the use of rEm18 (rEm18-ELISA) or a combination of rEm18 and natural Em2 antigen (Em2-Em18-ELISA) purified from *E. multilocularis* [37,60]. Additionally, rEm18-ELISA is currently the best serological monitoring test for assessing the curative course of AE [49]. Ahn et al. identified two proteoforms (6 and 8 KDa) as B3 antigen (EmAgB3), which exhibited AE-specific immune responses and emerged as a promising candidate for AE serological evaluation [50]. These proteoforms were separated from *E. multilocularis* hydatid fluid (EmHF) and have been proven to be highly relevant in assessing worm viability associated with the progression of AE. Korkmaz et al. [61] interpreted Em70 and Em90 as positive markers for AE. Serum samples from patients with AE exhibited strong reactivity, with 100% sensitivity at 70- and 90-kDa bands, and these bands were not recognized by sera from CE patients, with one exception. Özdemir et al. [62] conducted a preliminary exploration of serum-derived exosomal circular RNA as a potential biomarker for patients with hepatic alveolar echinococcosis. The serum level of platelet-derived growth factor-BB may serve as a simple, non-invasive, and rapid biomarker for evaluating the metabolic activity of lesions in patients with AE [63].

### 2.2. Molecular Techniques

Serological testing is a valuable tool for the early diagnosis of echinococcosis; however, its specificity and clinical effectiveness remain debatable. To date, no standardized diagnostic kit for echinococcosis has been universally accepted by clinicians. Some studies have reported the use of ELISA to detect Em antigen-specific antibodies in the urine of both mice and patients with AE [64]. While urine collection is simpler and safer than obtaining serum samples, there remains a scope for enhancing its sensitivity. In recent years, with the rapid advancement of molecular diagnostic techniques, metagenomic next-generation sequencing (mNGS) has gained significant attention as a promising approach for the etiological diagnosis of infectious diseases and may serve as a valuable tool for diagnosing AE [65]. Fan et al. [66] analyzed parasite-derived circulating cell-free DNA (cfDNA) in 149 plasma samples using a DNA sequencing-based method. Their study demonstrated that Em-cfDNA was detectable in the plasma of 100% of preoperative AE patients, while all non-AE patients and healthy volunteers tested negative. These findings suggest that Em-cfDNA levels have potential as a reference biomarker for assessing the therapeutic efficacy of surgical intervention in AE lesions. Polymerase chain reaction (PCR) analysis is frequently employed to identify the haplotype of *E. multilocularis*, aiding in tracing the origin of infection [67,68]. In human cases, PCR is mainly used for the direct detection of parasite nucleic acids in biological specimens, which are typically obtained through fine-needle aspiration biopsy.

### 2.3. Histology

The diagnostic process for AE typically involves clinical, radiological, and serological examinations, which are generally conducted before surgical intervention. However, a definitive diagnosis is usually based on histopathological analysis of surgically excised tissue, specially using periodic acid–Schiff (PAS-) staining [69]. PAS staining can specifically stain polysaccharides within the lesion tissues. In addition, the most commonly employed method for histological identification of AE is hematoxylin–eosin (H&E) staining. This technique allows for the visualization of key morphological and structural features of echinococcal cysts, including the laminated layer, germinal layers, protoscoleces, remnants of protoscolices with hooklets, and calcified corpuscles [70,71,72]. H&E staining can also reveal features such as central necrosis within the tissue, infiltration of neutrophils and eosinophils around the necrosis, the presence of non-caseating granulomas, and fibrous tissue formation [73]. The monoclonal antibody (mAb) Em2G11 was used as the primary antibody for the immunohistochemical analysis of the lesion tissues in vitro [74,75]. This antibody specifically targets the Em2 antigen. As shown in Figure 2A, during an immunohistochemical staining procedure using Em2G11 specific for metacestodes of *E. multilocularis* in human liver tissue, the antigen is detected within the laminated layer (two arrows, right) and in the necrotic area around the lesion (dashed lined area, right) [76]. Furthermore, immunohistochemical staining of a lymph node affected by the larval state of AE using the Em2G11 antibody reveals numerous small particles of *E. multilocularis* (spems) in both the follicle and perifollicular area (Figure 2B,C) [77]. Gomori methenamine silver (GMS) staining technique serves as an important auxiliary tool for the detection of parasitic diseases, facilitating the identification of parasites in samples such as tissues and feces. In H&E staining of liver tissue affected by AE, microcystic structures lined by a laminated grayish membrane are observed against a background of necrosis (Figure 2D). Subsequent staining with GMS and PAS further highlights the laminated membranes (Figure 2E,F) [78]. The graphic in Figure 2G shows histological analyses of liver tissues infected with AE, which are stained with PAS and characterized by five zones: zone 1: necrotic area, surrounding the laminated layer (a); zone 2: inner layer of epithelioid cells and granulocytes; zone 3: fibrotic area; zone 4: outer layer of lymphocytes; zone 5: adjacent liver tissue [79]. Additionally, Figure 2H presents an H&E-stained vertebral biopsy, revealing the parasite membrane (arrow) of the larval stage of *E. multilocularis* and the surrounding non-caseating granuloma (star) [80]. As illustrated in Figure 2I,J, the histology of an explanted liver exhibits the laminated layer of *E. multilocularis*, remnants of protoscolices, as well as the delicate germinal layer [81].

In a patient with isolated primary cerebral involvement, who had negative serological results and unremarkable chest and abdominal computed tomography findings, immunohistochemistry analysis confirmed the diagnosis of cerebral alveolar echinococcosis [82]. At this stage, histological analysis plays a particularly crucial role. Histological analysis revealed gliotic central nervous system tissue containing sharply demarcated fragments of a fibrous cyst wall, accompanied by marked inflammation and necrosis (Figure 3A). Under higher magnification, remnants of protoscolices with hooklets and calcified corpuscles were observed (Figure 3B,C). The multilaminar membranes showed strong PAS positivity (Figure 3D) and were immunolabeled with an antibody recognizing the Em2G11 protein (Figure 3E), further supporting the diagnosis of cerebral AE. In addition, Sulyok et al. [83] developed a deep learning-based decision support model for the histological diagnosis of AE, assisting pathologists in accurately classifying multilocular echinococcus liver lesions and normal liver tissues with high predictive performance.

### 2.4. Ultrasonography

Ultrasonography (US) has emerged as a pivotal imaging modality in the diagnosis and management of multilocular hydatid disease, particularly due to its non-invasive nature, accessibility, and cost-effectiveness [84]. This is especially beneficial in emergency settings or in remote areas where access to advanced imaging facilities may be limited. This is particularly important in populations requiring frequent imaging, as it minimizes cumulative radiation exposure. Ultrasound is the first choice for diagnosing and following up on hepatic alveolar echinococcosis (HAE) [85,86,87]. Ultrasound can intuitively provide diagnostic information such as the size, shape, boundary, and internal echo of the lesion of HAE. HAE usually presents as a solid nodular or giant mass with irregular shape and edge, unclear boundaries with liver tissue, no obvious capsule, uneven internal echo, and visible strong echoes or liquefied necrosis areas of varying sizes. Ultrasound plays a pivotal role in guiding punctures, such as ultrasound-guided hollow-core needle biopsy, a minimally invasive, reliable, and effective diagnostic tool used for confirming the diagnosis of HAE [88]. Deng and his colleagues have used ultrasound-guided percutaneous microwave ablation for treating HAE, which demonstrates a high degree of safety and efficacy [89].

Understanding the typical manifestations of HAE in diagnostic imaging may facilitate early diagnosis. Kratzer et al. [90] established a detailed sonomorphological classification based on a large sample of patients with confirmed hepatic AE, which is mainly divided into the following five patterns (Figure 4): (A) Hailstorm: ill-defined and irregular borders, uneven patterns, and high echogenicity formation; (B) Pseudocystic: a pseudocystic appearance resulting from central necrosis, surrounded by an irregular, ring-shaped area of high echogenicity; (C) Hemangioma-like: a relatively well-demarcated, heterogeneous tumor that appears hyperechoic compared to the surrounding liver parenchyma; (D) Ossification: isolated or clustered lesions, mostly with well-defined borders, accompanied by dorsal acoustic shadow; (E) Metastasis-like: mostly hypoechoic without the halo phenomenon, featuring a central, heterogeneous, hyperechoic scar. Sulima et al. [91] analyzed the ultrasound results of 58 patients with a possible or confirmed diagnosis of AE and found that the most common patterns of AE lesions in the liver were the Hailstorm and Pseudocystic patterns. They also found that there was no correlation between the clinical stage of the disease and the ultrasonographic appearance of lesions, and specific ultrasound patterns of lesions cannot determine the radicality of surgical treatment.

AE demonstrates infiltrative growth and less distinct boundaries on ultrasound imaging; it has a poorer prognosis due to its aggressive nature and tendency to be misdiagnosed as liver cancer. Advancements in ultrasound technology, such as contrast-enhanced ultrasound (CEUS), have further improved the diagnostic accuracy of echinococcosis [93]. CEUS using SonoVue (sulfur hexafluoride with a phospholipid shell; Bracco SpA, Milan, Italy), introduced in 2001, has been approved in many countries worldwide [94,95]. The size of these microbubbles is comparable to or smaller than that of red blood cells, which increases the blood echo rate and improves the signal-to-noise ratio in ultrasound imaging. Furthermore, CEUS enhances the visualization of blood vessel formation in liver lesions, allowing for more accurate qualitative assessment [96]. Global guidelines recommend CEUS for characterizing focal liver lesions and managing hepatocellular carcinoma [97,98]. Moreover, CEUS is as reliable as CT and MR in monitoring local ablation therapy for liver cancer, including the assessment of treatment response and follow-up investigations [99,100]. The research protocols for CEUS in the treatment of HAE are similar to those for focal liver lesions [101]. For instance, in alveolar echinococcosis, CEUS may show an irregular enhancement pattern with peripheral enhancement during the arterial phase, followed by inhomogeneous and low enhancement in the lesion margin during the peak intensity of the contrast agent. In addition, CEUS can visualize the parenchymal microvascular system, providing more information for differential diagnosis and assessment [102,103]. Previous studies have found that CEUS can visualize the microvascular status of the margin infiltration in HAE lesions, and the microvascular density directly reflects the biological activity of the lesions. Thus, CEUS has a high value in diagnosing the invasive proliferative activity at the edge of HAE lesions [104,105]. In CEUS images of hepatic AE, a circular rim-like enhancement belt is frequently observed. This enhanced strip may contain microvessels with richer blood supply and pathologically confirmed inflammatory reaction belt [106,107,108]. CEUS has better reliability than US in differentiating AE and CE; lesions with hyper-enhancement, heterogeneous or rim enhancement, and ill-defined boundary should be suspected as AE [109]. In addition, CEUS can accurately assess the activity of AE proliferation and infiltration areas, thereby enabling the evaluation of drug efficacy [110].

### 2.5. Computerized Tomography

Besides the US, computed tomography (CT) is presently the preferred diagnostic imaging among the available methods, as it is capable of detecting the highest number of lesions and clearly distinguishing characteristic calcifications [111,112]. Abdominal CT is more suitable for identifying calcifications and preoperative assessment. CT scan primarily relies on the determination of lesion size, rather than on the accurate assessment of AE proliferation and infiltration activity. In 2016, Graeter et al. [113] proposed five classifications for CT imaging of hepatic alveolar echinococcosis (Figure 5). Type I: diffuse infiltrating, Type II: primarily circumscribed tumor-like, Type III: primarily cystoid, Type IV: small-cystoid/metastatic, Type V: mainly calcified. After conducting a retrospective analysis of 228 patients with hepatic alveolar echinococcosis, they revealed that the most frequently encountered CT morphological pattern was the diffuse infiltration pattern (Type I), followed by the primarily circumscribed tumor-like appearances (Type II). The mainly calcified appearance (Type V) was observed less frequently. The *E. multilocularis* Ulm classification for computed tomography (EMUC-CT) offers the basis for the systematic description of the CT morphology of AE lesions of the liver [114]. A study has shown that there is a close correlation between the patterns of AE according to the EMUC-CT classification and the different histological patterns of AE infections [79]. The CT types mark consecutive stages of infection, which progress over time. The classification of calcifications in AE observed on CT is shown in Figure 6.

The “PNM” classification, developed by the Informal Working Group on Echinococcosis of the World Health Organization, serves as an international standard for assessing diagnostic performance and treatment outcomes. Using the PNM classification: P for parasitic mass in the hepatic artery, N for neighboring organ involvement, M for distant site involvement [115]. The initial signs and symptoms of liver involvement include abdominal pain, jaundice, hepatomegaly, weight loss, anemia, etc. As the disease progresses to its advanced stages, it can lead to portal hypertension, cholangitis, liver abscesses, and secondary biliary cirrhosis due to the invasion of biliary and vascular structures [116]. Graeter et al. [117] evaluated the vascular/biliary involvement and distant extrahepatic manifestations in 200 HAE patients from Germany, France, and China using the EMUC-CT classification. They found that different CT morphological patterns of hepatic AE lesions could affect the occurrence of vascular/biliary involvement and extrahepatic disease manifestations. CT scanning aids in assessing the relationship between vascular structures, biliary ducts, and liver lesions. It can also reveal extrahepatic organ involvement. This is crucial for determining the P, N, and M stages of the lesion and its resectability.

When suspected liver AE lesions are detected during contrast-enhanced CT scans, additional triphasic or non-enhanced CT scans are required to determine the presence of calcification and enhancement [118]. Consequently, imaging costs and exposure to ionizing radiation will increase. Dual-energy CT (DECT) and virtual non-enhanced (VNE) images can be utilized to create non-enhanced series from conventional contrast-enhanced images, significantly reducing the radiation dose [119]. Kantarcı et al. [118] conducted a study to investigate the diagnostic potential of VNE DECT reconstruction for hepatic AE. Their findings revealed that, despite notable decreases in noise, sharpness, and image quality in VNE images, the diagnostic performance for detecting hepatic AE was comparable to that of true non-enhanced (TNE) images. VNE images can effectively serve as a substitute for TNE images in diagnosing hepatic AE, thereby reducing exposure to ionizing radiation. A study has shown that three-phase helical CT can successfully visualize the vascularization associated with *E. multilocularis* lesions in the liver [120]. Angiogenesis occurs within the parasitic granuloma, which may participate in the transport of immune cells into and out of the lesion. Angiogenesis is regarded as an aspect of the immune response that occurs around the parasite [121]. Unlike traditional CT imaging, which can only reveal the morphology and density of pathological lesions, DECT is a novel functional imaging method that can enhance material differentiation and enable quantitative analysis [122]. The research conducted by Jiang [123] and colleagues demonstrated a positive relationship between the quantitative iodine concentration data derived from DECT and the microvascular density surrounding the lesions of AE. Consequently, DECT can be employed to visualize and quantify the vascularization of AE lesions, aiding in the determination of angiographic vascular characteristics, and it holds promise as a predictor of AE progression.

**Figure 6 diagnostics-15-00585-f006:**
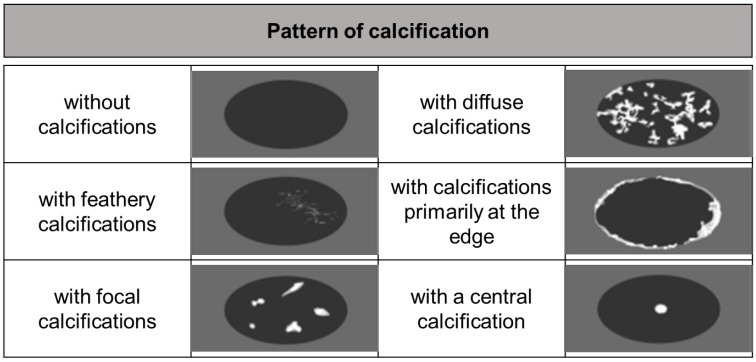
Patterns of calcification [124]. The calcification of AE in CT images is classified into five stages: no calcifications; feathery calcifications; focal calcifications; diffuse calcifications; calcifications primarily at the edge and central calcification.

Compared to ultrasonography, CT offers more objective and standardized procedures as well as automated operations [125]. It does not heavily rely on the proficiency of doctors. The image acquisition process is relatively simple and easy to operate, and the obtained images are relatively stable, providing great convenience for artificial intelligence (AI) in analyzing CT images. In recent years, the rapid development of AI, particularly radiomics within the field of radiology, has provided unprecedented opportunities for assessing HAE lesions. Radiomics performs high-throughput information extraction, reliably generating dependable and valuable data with greater consistency than visual observation. Wang et al. [126] developed an artificial intelligence system (EDAM: echinococcosis diagnostic AI system), to assist radiologists and clinicians in detecting and subtyping AE diseases through CT images. The sensitivity of EDAM is reliable across images from different CT manufacturers. EDAM outperforms most of the participating radiologists in detecting AE and CE. Xin et al. [127] employed a multi-scale feature convolutional neural network for automatic lesion segmentation and classification of hepatic echinococcosis (HE). The challenge of lesion segmentation is compounded by the low contrast among lesions, the liver, and other organs, as well as the interference from surrounding soft tissues, the impact of bone on calcified lesions, and noise in CT scans. Therefore, they first use the lesion region positioning module to locate the lesion and then employ the lesion region segmenting module for lesion segmentation. By leveraging the pathological features of HE to enhance abstract information and ignore detailed information, they obtain more accurate localization information, thereby achieving more precise image segmentation results. After segmenting the image, the results are classified using a convolutional neural network. The introduction of this network aids doctors in better locating lesions and obtaining more accurate diagnostic results. Deep learning is a machine learning technique rooted in artificial neural networks, capable of autonomous learning and extracting significant features from vast amounts of data, thereby finding widespread applications. The deep learning radiomics model developed by Jiang et al. has been used to explore potential applications in predicting the biological activity grading of hepatic echinococcosis based on CT images [128,129].

### 2.6. Magnetic Resonance Imaging

When meeting uncertain cases with noncalcified or partially calcified lesions, MRI may facilitate displaying the characteristic multivesicular structure, necrotic areas, and proximity to vascular structures. Compared to CT, magnetic resonance imaging (MRI) offers the advantage of being non-radiative, particularly excelling in imaging soft tissue structures by providing high-resolution and clear images. MRI utilizes diverse scanning techniques, including T1, T2, and diffusion-weighted imaging (DWI), to furnish a holistic diagnostic perspective. These methods each contribute distinct information about tissue characteristics and disease manifestations, thereby enriching the diagnostic detail. Accordingly, MRI demonstrates superior tissue contrast in depicting small vesicle structures of lesions and showing proximity to vascular structures, facilitating early and non-invasive detection of small lesions [86,130]. Compared to other non-invasive imaging techniques, MRI provides superior visualization of mass contours, central necrosis, vascular relationships, and extrahepatic extensions of AE. Fibrosis and parasitic tissue exhibit low signal intensity on T1-weighted (T1W) images, often resulting in better edge definition and clarity. T2-weighted images are suitable for detecting small cystic peripheral extensions and defining the central necrosis zone [131,132]. The diagnosis of small peripheral cysts is important because they are considered to be active components of the lesion. The cystic components consist of metacestode vesicles and liquefied necrosis, characterized by low T1-weighted and high T2-weighted MRI signal intensity. The metacestode vesicles manifest as small, round cysts, while liquefied necrosis presents as large and/or irregular cysts. The solid components, featuring low T1-weighted and low to intermediate T2-weighted MRI signal intensity, encompass coagulative necrosis, granulomas, and/or calcification, reflecting the chronic fibrotic inflammatory response of the host [133]. On postgadolinium images, lesions did not reveal enhancement [134]. Solid lesions and mixed-type lesions exhibit some similarities in their disease courses and are the primary forms of advanced AE. Pseudocystic lesions represent neither the early nor the advanced stage of AE; rather, they are a specific manifestation during the progression of AE. MRI is the optimal imaging technique for characterizing echinococcosis lesions and determining their intrahepatic and extrahepatic extensions, particularly to the subdiaphragmatic region and hepatic hilum. It best demonstrates the multivesicular cystic structure of AE lesions, which are depicted as areas of high T2 signal intensity [135].

In 2003, Kodama and colleagues classified the MRI imaging results of patients with AE (Figure 7) [133]. They classified AE into five types based on its imaging manifestations on MRI: Type 1 consists of multiple small round cysts without a solid component; Type 2 features multiple small round cysts with a solid component; Type 3 is characterized by a solid component surrounding large and/or irregular cysts with multiple small round cysts; Type 4 comprises a solid component without cysts; and Type 5 is a large cyst without a solid component. The reliability of Kodama’s classification is excellent and has received widespread recognition, allowing for the identification of five types of liver lesions based on morphological features on T2-weighted MRI sequences. Brumpt et al. further refined the Kodama classification by subdividing Type 3 into IIIa with microcysts and IIIb without, based on the crucial role of microcysts in therapeutic decision-making for AE lesions [136]. T2-weighted MRI imaging exhibits better imaging performance in the early diagnosis of AE, compared to CT or US. In a 74-year-old male patient, seven years after resection of rectal cancer, CT scans and ultrasound images revealed two small, uniform low-density and hypoechoic lesions in the liver, respectively. These lesions were later histologically confirmed as early but active AE. Abdominal MRI, on T2-weighted images, demonstrated two isointense tumor lesions with small areas of high-signal intensity at their centers. The MRI findings seemed to accurately reflect both the macroscopic and microscopic manifestations of early AE, with each liver lesion containing an active cyst at its center [137].

The assessment of the metacestode activity status using 18-fluoro-deoxyglucose positron emission tomography combined with computed tomography (PET/CT) is widely accepted and recognized. Azizi et al. [138] correlated the presence of AE liver lesions defined by Kodama et al. on MRI with metabolic activity visualized on PET/CT and found that the presence of microcysts in AE liver lesions detected by MRI is associated with metabolically active disease. Their study revealed that all Type 1 and the majority of Type 2 (90.9%) and Type 3 (87.5%) lesions exhibited increased perilesional FDG uptake on PET/CT images. In fact, all three types are composed of multiple small cysts, which directly reflect the infracentimetric parasitic vesicles. These small cysts cluster around the periphery of the lesion in a “bunch of grapes” or a “honeycomb-like” patterns, which are considered to represent the early stages of the disease. In fact, the correlation between the microvesicle images on MRI and positive PET/CT is likely explained by more numerous and metabolically active immune cells within the lesion areas where active vesicles are present [139]. A study has shown that imaging features based on T2-weighted imaging (T2WI) and machine learning models can be used to assess the biological activity of HAE lesions, which is conducive to the selection and monitoring of clinical treatment methods [140]. Susceptibility-weighted imaging (SWI) is an MRI technique that may assist in identifying calcified regions and has recently been proven to detect liver calcification, potentially reducing the need for CT imaging in assessing echinococcosis [141]. Intravoxel incoherent motion diffusion-weighted magnetic resonance imaging (IVIM DW MRI) was compared with T1 mapping for the characterization of HAE, revealing that IVIM-derived parameters, especially the perfusion fraction (f), provided better differentiation of solid HAE components from background liver parenchyma than T1 relaxation times [142].

Radical resection of hepatic echinococcosis is an effective treatment option, but it faces the challenge of the parasite’s invasive growth and often infiltrating intrahepatic blood vessels [143]. Therefore, accurate preoperative assessment of the patient’s vascular invasion status is pivotal to the success of the surgical procedure [144]. MRI is a crucial imaging technique capable of revealing the characteristic structures of blood vessels and bile ducts in HAE, as well as their invasive status. It holds significant value in analyzing the features of vascular infiltration and the growth patterns of lesions [145]. A study presents an experimental study of synchrotron-based X-ray propagation-based phase-contrast computed micro-tomography (PPCT) combined with generalized phase and attenuation duality (PAD) phase retrieval, aiming to quantitatively visualize tiny density variations in soft tissues and organs of rat models infected with hepatic echinococcosis at different stages [146]. The results demonstrate the superiority of PAD-PPCT in clearly observing the infiltration of tiny vesicles in HE lesions compared to common medical imaging techniques such as MRI, CT, and ultrasound. They also provide evidence of the early invasion of HAE into liver tissue and its spread through blood flow systems with abundant blood supply. Establishing AE disease models in rats or mice, followed by MRI analysis at various stages of disease growth, facilitates insights into the early stages of the disease and its dynamic progression over different periods [147,148].

DWI relies on the random microscopic motion of water protons to qualitatively and quantitatively assess tissue diffusivity through the apparent diffusion coefficient (ADC). Diffusion is inversely related to cellularity, cell membrane integrity, and lipophilicity. MRI examinations using standard sequences are insufficient for diagnosing hepatic AE lesions, and ADC can serve as a supplementary tool to traditional MRI [149]. Becce et al. [150] reported diffusion-weighted MRI findings in HAE and evaluated the potential role of the ADC in characterizing the lesion features. The results indicated that ADC values were significantly lower in liver AE lesions, particularly in purely cystic lesions. ADC measurements may assist in differentiating AE liver lesions from simple biliary cysts, thereby narrowing the scope of differential diagnosis [30]. In addition, the ADC value aids in differentiating AE from hepatocellular carcinoma and intrahepatic cholangiocarcinoma [151]. The ADC value of HAE is higher than that of malignant liver lesions but lower than that of other cystic lesions within the liver. In suspected cases of HAE, DWI serves as a useful adjunctive test to conventional liver MRI, and it also provides potentially valuable information regarding the activity/vitality of HAE [152].

### 2.7. Positron Emission Tomography

Glucose is the primary energy substrate for parasites and fluorodeoxyglucose positron emission tomography (FDG-PET) has been the gold standard for assessment of the metacestode activity status, which is essential for the individual treatment strategy of the AE patient [104,153,154]. FDG-PET was introduced 25 years ago to monitor the progression of AE lesions. Positive FDG uptake indicates active lesions, while negative FDG uptake suggests parasite abortion, an indication of albendazole withdrawal [155]. This method has proven to be effective, but there are cases of recurrence after discontinuation of medication in some patients [156]. Currently, the improved PET imaging protocol requires both the 3 h post-FDG injection delayed image and the 1 h post-injection image to be negative in order to determine that the result is negative [153]. Delayed imaging can reduce false negatives and enhance the sensitivity in determining the metabolic activity of AE (Figure 8).

As shown in Figure 9, in vitro experiments have demonstrated that immune cells surrounding the AE lesions exhibit high uptake of FDG, whereas parasitic cells within the cysts show low uptake of FDG [157,158,159,160]. Therefore, FDG-PET can indirectly assess parasite activity by monitoring the activity of host immune cells [161]. The metabolic hyperactivity of AE liver lesions on FDG-PET/CT is strongly correlated to the presence of microcalcifications on CT but not with the presence of macrocalcifications [84]. The absence of microcysts on MRI is closely associated with metabolically inactive diseases [138].

Graeter et al. [124] firstly correlated the standardized uptake value (SUV) of HAE on FDG-PET with the morphological features and EMUC-CT classifications of the lesions. Their results show that the SUVs were increased for lesions with EMUC-CT types I–IV primary morphology, compared to the surrounding healthy liver tissue (SUV = 2.5 ± 0.4; *p* < 0.05). The SUVs of Types I and III were the highest, while those of Type IV were the lowest. Type IV had the highest rate of negative FDG-PET findings, with statistically significant differences compared to the other types, indicating that these SUVs might reflect different stages of the disease. Due to the small number of cases, it was not possible to evaluate Type V lesions. Furthermore, the FDG uptake differed significantly between cases with positive and negative Em2+ serology, in which Type IV lesions are typically associated with negative Em2+ serology. The metabolic rate of FDG images generated with traditional and relative Patlak analysis demonstrates superior performance in visualizing HAE compared to static SUV images [163]. Ozmen et al. [164] retrospectively analyzed 36 patients with pulmonary hydatid disease who underwent FDG-PET/CT imaging, examining lesion characteristics, SUVmax and HUmean values, and lymphatic FDG uptake to assess disease complexity. Their study found that higher SUVmax values may indicate complicated pulmonary hydatid disease, and FDG-PET can aid in prioritizing surgical intervention and evaluating treatment response. For advanced HAE cases where radical hepatectomy is not feasible, autologous liver transplantation (ALT) has emerged as an effective treatment option in recent years. However, various factors post-ALT can readily lead to recurrence and distant metastasis of HAE. ^18^F-FDG PET/CT is considered to play a crucial role in assisting early detection and assessment of liver function status post-ALT, as well as in evaluating the risk of recurrence or metastasis [165]. Aini et al. [166] use PET/CT and multi-site sampling methods to quantitatively evaluate the range and metabolic activity of HAE lesion microenvironment and connect it with immunological and pathophysiological changes, providing a reference for surgical resection of the lesions and more accurate sample acquisition.

FDG-PET is also used for long-term follow-up of AE (Figure 10). The absence of metabolic activity indicates suppression of parasite activity, which is not equivalent to parasite death. This suppression can persist for several years, and oral medication therapy should be reinitiated when recurrence is detected by PET [167,168,169]. In a retrospective study of 179 AE patients who underwent PET/CT scans, it was found that as the clinical status progressed, significant changes occurred in total immunoglobulin E (IgE), parasite-specific IgE, and serological status (using crude antigen preparations or recombinant/purified antigens such as EM10, Em18, and Em2) [170]. Notably, these serological biomarkers were also significantly higher in patients with positive PET results. Multiple studies have shown that combining FDG-PET with serological testing can further improve the accuracy of parasite activity identification [171]. For patients with AE who are unable to undergo surgical resection, oral albendazole treatment is often prescribed; the treatment usually is life-long. For patients with inactive disease, a structured treatment interruption (STI) of drug therapy may be a goal, not only to save costs but also to improve quality of life. Ammann et al. [172] evaluated FDG-PET/CT and antibody levels against recombinant Emll/3-10 antigen as markers of parasite vitality, serving as a reliable tool to allow for the selection of patients who can safely discontinue chemotherapy with a low risk of AE recurrence. Husmann et al. [173] hold the view that negative FDG-PET/CT results combined with no detectable levels of Em-18 antibodies may allow for the safe discontinuation of benzimidazole therapy in patients with AE. The quantitative imaging parameter SUVratio, obtained through PET/CT, correlates with the time to reach no detectable levels of Em-18 antibodies and the duration of benzimidazole treatment. In patients presenting negative results for both indicators, a watch-and-wait strategy might be permissible [174].

Although FDG-PET is currently the standard non-invasive tool for detecting metabolic activity in HAE lesions, the complexity and high cost of the examination process mean that it cannot be used routinely. Its high uptake correlates with the high metabolic activity of local immune cells and is not specific to echinococcosis as an imaging agent. This may make it difficult to distinguish HAE from other diseases with high FDG uptake, such as infections and inflammation [175]. The latest developments in whole-body PET/MRI imaging and its introduction into clinical routines offer the potential for an alternative dual imaging technique that combines the high-resolution anatomical imaging of PET with MRI. This may enhance soft tissue differentiation, assess the vitality of lesions, and consequently improve the guidance for therapeutic management. In accordance with the requirements for long-term follow-up of AE, PET/MRI has the potential to significantly reduce patients’ radiation exposure by replacing CT with MRI imaging [162,176].

## 3. Discussion

In the intricate diagnostic landscape of AE, a devastating parasitic disease primarily affecting the liver, a comprehensive diagnostic approach integrating serological and imaging techniques is indispensable. Serological testing, often serving as the initial screening tool, harnesses the patient’s immune response to detect specific antibodies against *E. multilocularis*. Tests like the indirect hemagglutination test (IHA) and Western blot, despite their varying sensitivities and specificities, play a crucial role in endemic areas, where early detection can significantly improve patient outcomes. However, the interpretation of serological results must be approached with caution due to potential cross-reactivity with other helminthic infections and the possibility of false negatives, particularly in early-stage or immunocompromised patients. Thus, serological testing should be viewed as a first step rather than a definitive diagnosis.

Imaging technologies further elevate the diagnostic precision of AE, with the US serving as a primary, cost-effective, and widely accessible modality. US can identify characteristic hepatic lesions with a ‘bubble-like’ or ‘honeycomb’ appearance, which are hallmarks of AE. However, for a deeper and more detailed understanding of the disease, advanced imaging techniques such as CT, MRI, and PET offer supplementary insights that greatly increase our comprehension. CT scanning provides high-resolution anatomical visualization of lesion size, location, calcification patterns, and adjacent organ involvement, aiding in accurate staging and surgical planning. MRI, with its exceptional soft tissue contrast resolution, detects subtle changes in tissue texture, perfusion abnormalities, and even early infiltrative growth patterns that may precede gross structural alterations, thereby enhancing early detection and assessment of disease progression. PET scanning, leveraging the metabolic activity of the parasite, adds another dimension by highlighting areas of active parasite metabolism, which can be crucial for guiding therapeutic interventions, monitoring treatment response, and assessing recurrence. By synergizing these imaging modalities with serological testing, healthcare providers can formulate a robust diagnostic strategy that significantly enhances our ability to detect, stage, monitor, and ultimately manage alveolar echinococcosis, ultimately improving patient care and outcomes.

For each imaging method (US, CT, MRI, PET), AE exhibit distinct imaging features compared to cancers such as hepatocellular carcinoma (HCC) or cholangiocellular carcinoma (CCC). On ultrasound, HCC and CCC appear as hypoechoic masses with vascularity, whereas AE presents as an irregular, hypoechoic, infiltrative lesion with a “honeycomb” structure and minimal vascularity. On CT, HCC shows arterial phase hyperenhancement and washout; CCC demonstrates delayed enhancement; while AE appears as a multi-lobulated, infiltrative lesion with calcifications and little contrast enhancement. On MRI, HCC and CCC typically show hyperintensity on T2, with enhancement patterns differentiating them, while AE appears hypointense on both T1 and T2, with irregular peripheral enhancement but no washout. On PET, AE is distinguished by intense peripheral FDG uptake surrounding necrotic or calcified areas, unlike the variable uptake in HCC and the typically high uptake in CCC. These imaging characteristics are key for distinguishing AE from liver malignancies.

## 4. Conclusions

The continuous evolution of diagnostic methodologies for HAE, a devastating parasitic infection, underscores the critical need for precise and early identification to optimize patient management and outcomes. This review encompasses the latest developments in serological, histological, and imaging techniques, including ultrasound, CT, MRI, and PET/CT, etc. Serologically, the integration of refined antigens like Em2/Em18 and combined natural and synthetic antigen Em2-Em18-ELISA has significantly enhanced the diagnostic accuracy of HAE. These advanced serological tests not only exhibit high sensitivity and specificity but also facilitate early detection, which is crucial for initiating timely and effective treatment interventions. Histologically, advancements in tissue biopsy techniques and immunohistochemical staining have enabled a more definitive diagnosis of HAE by identifying characteristic lesions and parasite structures within liver tissues. These techniques are particularly valuable in complex cases where serological or imaging findings are inconclusive. Imaging modalities have undergone substantial advancements, significantly contributing to the non-invasive diagnosis of HAE. Ultrasound remains a primary screening tool due to its wide availability and low cost, while CT and MRI offer superior resolution and detailed anatomical visualization, aiding in the assessment of lesion size, location, and involvement of adjacent structures. Notably, PET/CT has emerged as a promising tool for evaluating metabolic activity within lesions, providing insights into disease progression and response to treatment. Collectively, these diagnostic advancements underscore the importance of a multidisciplinary approach, combining serological, histological, and imaging techniques to establish a comprehensive diagnostic strategy for HAE. Early and accurate diagnosis, enabled by these cutting-edge methods, is pivotal in guiding appropriate treatment strategies, improving patient outcomes, and ultimately reducing the morbidity and mortality associated with this challenging disease.

## 5. Future Directions

Despite the significant strides made in the diagnostic methodologies for HAE, there remains a pressing need for continued innovation and refinement to further enhance the accuracy, efficiency, and accessibility of these techniques. Future directions in the field should focus on several key areas. One priority is the development of novel serological markers with even higher sensitivity and specificity. This could involve the identification of new parasite antigens or the refinement of existing diagnostic platforms, such as ELISA, to improve their diagnostic performance. Additionally, the exploration of serum proteomics and metabolomics may uncover novel biomarkers that could complement or surpass current serological tests.

In the realm of imaging, advancements in AI and machine learning algorithms hold great promise for improving the interpretation of diagnostic images. These technologies have the potential to automate the detection and characterization of HAE lesions, reducing the reliance on human interpretation and enhancing diagnostic consistency. Furthermore, the integration of AI with advanced imaging modalities, such as PET/CT, could provide more nuanced insights into disease activity and treatment response. Another area of focus should be the enhancement of histological diagnostic techniques, particularly through the development of novel staining methods and immunohistochemical markers that can differentiate HAE from other liver pathologies with overlapping histological features. Lastly, collaborative research efforts across disciplines and international borders are essential to accelerate the translation of promising diagnostic innovations from the bench to the bedside. By fostering a collaborative environment, researchers can pool their expertise, resources, and patient samples to validate new diagnostic tools and drive progress in the field.

In conclusion, the future of HAE diagnosis is poised for substantial advancements through the pursuit of novel serological markers, AI-driven imaging innovations, refined histological techniques, and interdisciplinary collaboration. These efforts will undoubtedly lead to more accurate, efficient, and accessible diagnostic strategies, ultimately improving the care and outcomes for patients with this challenging disease.

## Figures and Tables

**Figure 1 diagnostics-15-00585-f001:**
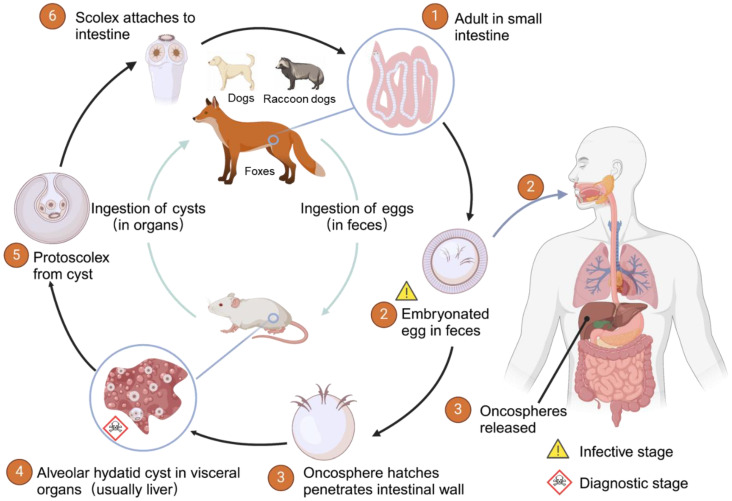
The life cycle of *Echinococcus multilocularis*. Foxes serve as the primary definitive hosts, and other hosts include canines such as dogs and raccoon dogs. Rodents such as small rodents serve as typical intermediate hosts. Human is infected as an aberrant intermediate host. Created with BioRender.com.

**Figure 2 diagnostics-15-00585-f002:**
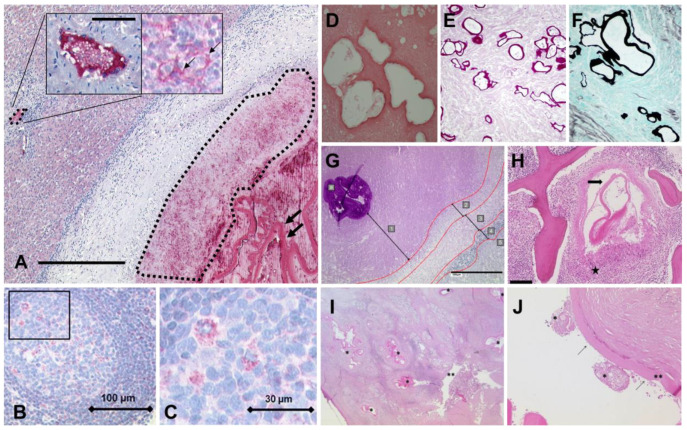
*E. multilocularis* lesion in human liver tissue. Bar = 750 µm; bar insert = 40 µm (**A**); arrow (right): laminated layer; arrow (in the insert): specific staining in lymphoid tissue of a regional lymph node on the surface of cells [76]. Immunohistochemical staining with the Em2G11 antibody detects numerous spems in the follicle and the perifollicular area of a lymph node with *E. multilocularis* (spems) involvement (**B**,**C**) [77]. Hematoxylin–eosin (H&E) staining shows microcystic structures lined by a laminated grayish membrane on a background of necrosis (**D**), and staining with Gomori methenamine silver (GMS) and Periodic acid−Schiff (PAS) further highlights the laminated membranes (**E**,**F**) [78]. Examples of histological analyses of liver alveolar echinococcosis infections. Sections were stained with PAS (**G**) [79]. Parasite membrane (arrow) of the larval stage of *E. multilocularis*, with surrounding non caseating granuloma (star). Scale bar = 200 µM (**H**) [80]. The histology of the explanted liver showing the laminated layer of *E. multilocularis* (*) in close proximity to a large bile duct (**) in the hepatic hilus region (**I**). Magnification of a metacestode. Remnants of two protoscolices (*), the laminated layer (**), and the delicate germinal layer (arrows) are shown (**J**) [81].

**Figure 3 diagnostics-15-00585-f003:**
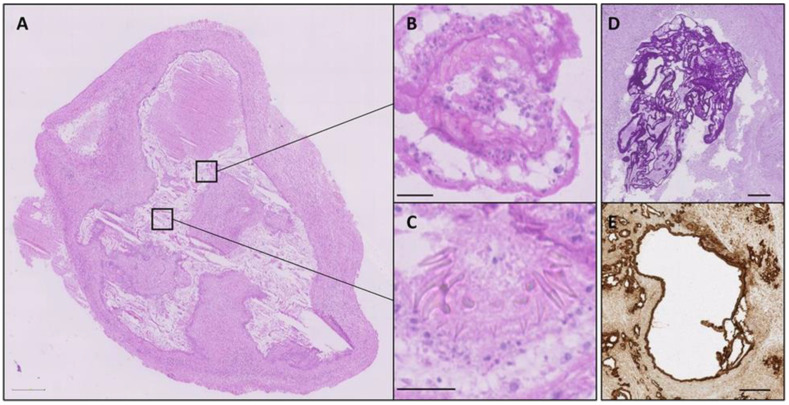
H&E staining shows overview section (**A**) with marked areas magnified (**B**,**C**). PAS stain (**D**) and Em2G11 immunohistochemistry (**E**). Scale bars = 1 mm (**A**), 20 μm (**B**,**C**), and 100 μm (**D**,**E**) [82].

**Figure 4 diagnostics-15-00585-f004:**
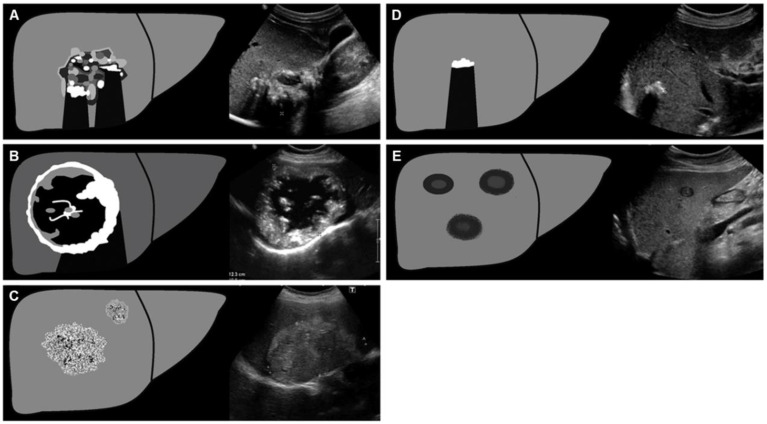
The detailed sonomorphological classification proposed by Kratzer et al. [90] and reprinted by Yangdan et al. [92] (**A**) Hailstorm; (**B**) Pseudocystic; (**C**) Hemangioma-like; (**D**) Ossification; (**E**) Metastasis-like.

**Figure 5 diagnostics-15-00585-f005:**
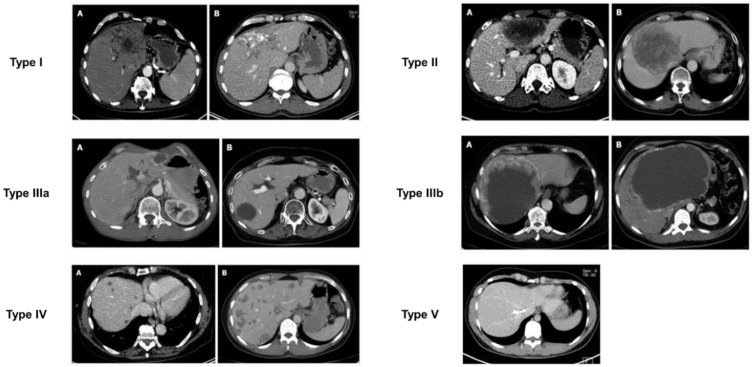
The EMUC-CT classification system, based on Graeter et al. [113] Type I: diffuse infiltrating with cystoid portion (**A**) and without cystoid portion (**B**). Type II: primarily circumscribed tumor-like with cystoid portion (**A**) and without cystoid portion (**B**). Type IIIa: primarily cystoid, intermediate (approximately 3–8 cm), with more solid portions at the edge (**A**) and without more solid portions at the edge (B). Type IIIb: primarily cystoid, widespread (approximately >8 cm), with more solid portions at the edge (A) and without more solid portions at the edge (**B**). Type IV: small cystoid/metastatic (approximately <3 cm), exclusive occurrence of the central calcification. Type V: mainly calcified.

**Figure 7 diagnostics-15-00585-f007:**
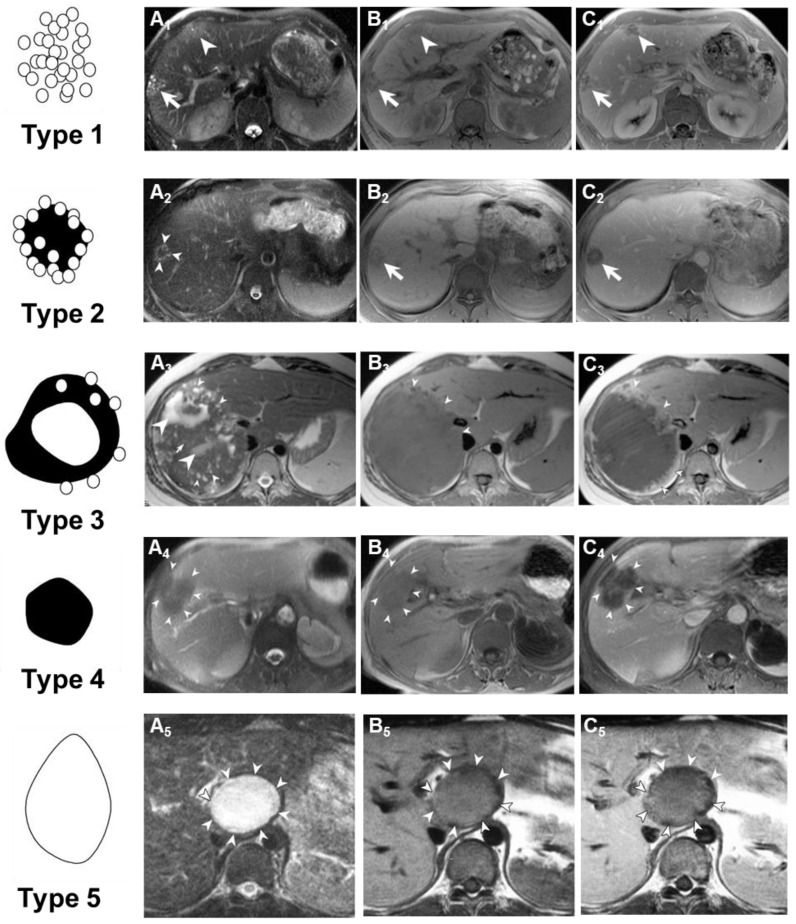
Schematic shows classification scheme of MR findings of alveolar echinococcosis in the liver by Kodama et al. [133] Type 1: multiple small round cysts without a solid component; Type 2: multiple small round cysts with a solid component; Type 3: a solid component surrounding large and/or irregular cysts with multiple small round cysts; Type 4: a solid component without cysts; Type 5: a large cyst without a solid component. Transverse MR images show Type 1–5 lesions. T2-weighted images (**A**), T1-weighted images (**B**), and contrast-enhanced T1-weighted images (**C**) obtained from MRI and shown the lesions (arrowheads). These arrows point to the diseased tissue.

**Figure 8 diagnostics-15-00585-f008:**
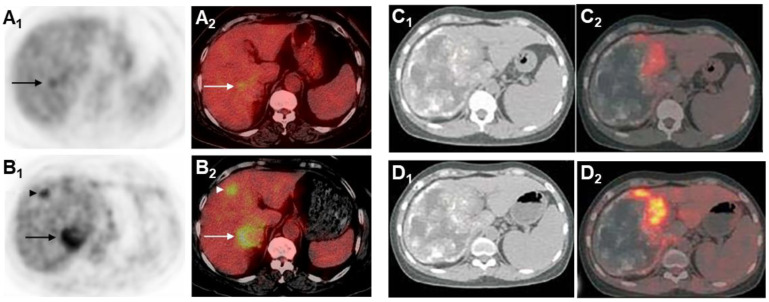
Standard PET acquisition (**A_1_**) and PET/CT acquisition (**A_2_**). Delayed PET acquisition (**B_1_**) and PET/CT acquisition (**B_2_**) [153]. The images acquired through delayed acquisition are clearer, showing a lesion that is visible only in delayed acquisition. Early image acquisition at 1 h after FDG injection (**C_1_**,**C_2_**) and delayed image acquisition at 3 h after FDG injection (**D_1_**,**D_2_**) significantly enhanced images of FDG uptake, with an increase in SUV value [34].

**Figure 9 diagnostics-15-00585-f009:**
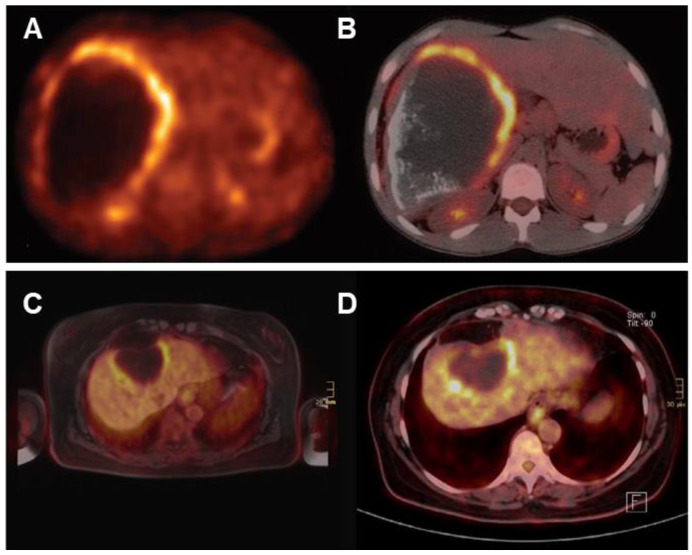
A doughnut sign of hepatic alveolar echinococcosis on FDG PET/CT (**A**,**B**) [154]. FDG-PET/MRI (**C**) and FDG-PET/CT (**D**) scan of a patient with alveolar echinococcosis [162]. The characteristic metabolic appearance is associated with peripheral hypermetabolic rims induced by inflammation and central hypometabolic areas resulting from necrosis.

**Figure 10 diagnostics-15-00585-f010:**
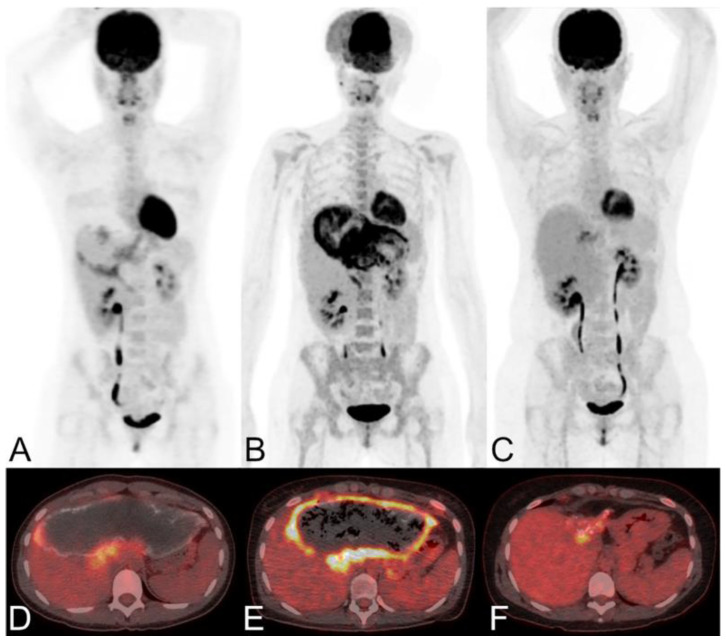
Long-term dynamic PET/CT monitoring of a female patient with alveolar echinococcosis exhibiting large liver manifestations [171]. The patient was 24 years old in 2005 (maximum intensity reconstructions of PET (**A**) and fused PET/CT images (**D**)), 32 years old in 2012 (**B**,**E**), and 40 years old in 2020 (**C**,**F**). She has been treated with albendazole since 1997, and therapy was ongoing at the last clinical follow-up in 2021.

## Data Availability

No new data were created or analyzed in this study.

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
