# Peer review of "Advances in Novel Diagnostic Techniques for Alveolar Echinococcosis"

_diagnostics, 2025, doi:10.3390/diagnostics15050585_

Round 1

Reviewer 1 Report

Comments and Suggestions for Authors

Your work is important on the merit of the understanding and knowledge of Alveolar Echinococcosis diagnosis, although there are different points that must first be addressed.

I would advise you to review the manuscript; a native English should also review the manuscript before the next submission.

I strongly advise you to consult the rules for authors; a template is available on the website that should be used in the submission, facilitating the review of the article. Ex. The side numbering is missing, very helpful in the review and response from the authors.

Overall, the article has a lack of connection between the different parts. From my point of view, it seems that it would gain a lot if, when describing the different diagnostic techniques, you could somehow interconnect them. You mentioned at the abstract these connection, but I did not read throughout the text. The information included in the different sections, as well as in its subtitles, and other sections, are often redundant and confuse. Ex. At the section of 2. Diagnostic, subsection 2.1 Serology you mentioned DNA techniques, such as PCR or NGS, I think it will be a gain a Molecular Techniques subtitle of Diagnostic.

In my point of view, your article to be accepted for publication, should be restructured to be more comprehensive and focused more on an innovative nature, such as new technologies for diagnosis.

I sincerely hope that in the future I can review your article with the necessary changes.

Comments on the Quality of English Language

I would advise you to review the manuscript; a native English should also review the manuscript before the next submission.

Author Response

Thank you very much for taking the time to review this manuscript. Please find the detailed responses below and the corresponding revisions are highlighted in red in the revised manuscript.

Point-by-point response

Reviewer 1#

(1) Your work is important on the merit of the understanding and knowledge of Alveolar Echinococcosis diagnosis, although there are different points that must first be addressed. I would advise you to review the manuscript; a native English should also review the manuscript before the next submission.

Re: Thank you for recognizing the importance of our work in Alveolar Echinococcosis diagnosis. We acknowledge your concern and have already arranged for a native English speaker to review and improve the manuscript.

(2) I strongly advise you to consult the rules for authors; a template is available on the website that should be used in the submission, facilitating the review of the article. Ex. The side numbering is missing, very helpful in the review and response from the authors.

Re: Thank you for your helpful suggestion. We have consulted the author's guidelines and ensured the manuscript follows the required format, including adding side numbering to facilitate the review process.

(3) Overall, the article has a lack of connection between the different parts. From my point of view, it seems that it would gain a lot if, when describing the different diagnostic techniques, you could somehow interconnect them. You mentioned at the abstract these connections, but I did not read throughout the text. The information included in the different sections, as well as in its subtitles, and other sections, are often redundant and confuse. Ex. At the section of 2. Diagnostic, subsection 2.1 Serology you mentioned DNA techniques, such as PCR or NGS, I think it will be a gain a Molecular Techniques subtitle of Diagnostic.

Re: Thank you for your insightful feedback. We have made efforts to improve the connections between sections and tried to reduce redundancy where possible. Additionally, we have added a separate subtitle for Molecular Techniques under the Diagnostic section, as suggested. Page 4, line 136.

(4) In my point of view, your article to be accepted for publication, should be restructured to be more comprehensive and focused more on an innovative nature, such as new technologies for diagnosis. I sincerely hope that in the future I can review your article with the necessary changes.

Re: Thank you for your insightful comments and suggestions. I truly appreciate your thoughtful feedback, especially regarding the focus on innovative diagnostic technologies. However, as the research on novel diagnostic methods for AE is still in its early stages, there is currently limited material to fully restructure the article around this. It seems more appropriate to incorporate these new technologies into each diagnostic method section.

Reviewer 2 Report

Comments and Suggestions for Authors

The review “Advances in Novel Diagnostic Techniques for Alveolar Echinococcosis“ is interesting and gives a good overview.

Minor comments/questions:

1) For each imaging method (US, CT, MRI, PET), it would be good to include a paragraph on the difference to important diagnoses that need to be ruled out (especially cancers such as hepatocellular carcinoma or cholangiocellular carcinoma).  

2) For PET/CT, how long should the measurement period be? Is it extended 3h? And how often and for how long should PET/CT be performed in the course of HAE over the years?

3) Can you briefly explain the significance of a diagnostic liver biopsy?

4) What is the importance of interdisciplinary diagnosis, for example in a multidisciplinary board?

5) When is it a confirmed E. multilocularis infection?

Line 45: Echinococcosis in Austria is no longer restricted to the west, but is now widespread throughout the country. (Schneider R, Aspöck H, Auer H. Unexpected increase of alveolar echinococcosis, Austria, 2011. Emerg Infect Dis. 2013 Mar;19(3):475-7. doi: 10.3201/eid1903.120595)

Line 64: Echinococcus multilocularis is not in italics. 

Line 83: In Figure 1 foxes and coyotes are listed as the most important final hosts, in the text foxes, canines such as dogs and raccoons (?), please standardise text and figure.

Author Response

Thank you very much for taking the time to review this manuscript. Please find the detailed responses below and the corresponding revisions are highlighted in red in the revised manuscript.

Point-by-point response

Reviewer 2#

The review “Advances in Novel Diagnostic Techniques for Alveolar Echinococcosis “is interesting and gives a good overview. Minor comments/questions:

(1) For each imaging method (US, CT, MRI, PET), it would be good to include a paragraph on the difference to important diagnoses that need to be ruled out (especially cancers such as hepatocellular carcinoma or cholangiocellular carcinoma).

Re: Thank you for your valuable suggestion. We have added a paragraph that highlights the differences of these techniques in ruling out important differential diagnoses, particularly hepatocellular carcinoma (HCC) and cholangiocellular carcinoma (CCC). Page 17, line 637−649.

For each imaging method (US, CT, MRI, PET), AE exhibits distinct imaging features compared to cancers such as hepatocellular carcinoma (HCC) or cholangiocellular carcinoma (CCC). On ultrasound, HCC and CCA appear as hypoechoic masses with vascularity, whereas AE presents as an irregular, hypoechoic, infiltrative lesion with a "honeycomb" structure and minimal vascularity. On CT, HCC shows arterial-phase hyperenhancement and washout, CCA demonstrates delayed enhancement, while AE appears as a multi-lobulated, infiltrative lesion with calcifications and little contrast enhancement. On MRI, HCC and CCA typically show hyperintensity on T2, with enhancement patterns differentiating them, while AE appears hypointense on both T1 and T2, with irregular peripheral enhancement but no washout. On PET, AE is distinguished by intense peripheral FDG uptake surrounding necrotic or calcified areas, unlike the variable uptake in HCC and the typically high uptake in CCA. These imaging characteristics are key for distinguishing AE from liver malignancies.

(2) For PET/CT, how long should the measurement period be? Is it extended 3h? And how often and for how long should PET/CT be performed in the course of HAE over the years?

Re: PET/CT plays an important role in the diagnosis and evaluation of AE, especially in assessing the activity and extent of the disease. The imaging is generally performed 1 hour after FDG injection, though delayed imaging at 3 hours post-injection can also be helpful in improving sensitivity and reducing false negatives.

Regular PET/CT scans play a crucial role in disease monitoring during treatment and follow-up after therapy. When used for treatment monitoring, the frequency of scans is typically determined by the treatment regimen and the patient’s specific condition, which are generally performed every 3 to 6 months. Positive FDG uptake indicates active lesions, while negative uptake suggests parasite abortion and possible albendazole withdrawal. Though effective, this approach may still lead to recurrence after discontinuation. The improved PET protocol requires both 1-hour and 3-hour post-FDG injection images to be negative for a definitive result. Delayed imaging enhances sensitivity and reduces false negatives in assessing AE metabolic activity.

For post-treatment follow-up, scans are typically conducted every 6 to 12 months in the early stages, with the interval gradually extended based on disease stability to monitor for recurrence or progression.

(3) Can you briefly explain the significance of a diagnostic liver biopsy?

Re: Thank you for your valuable comment regarding the significance of a diagnostic liver biopsy. A liver biopsy provides definitive histopathological evidence of AE by identifying parasitic structures, granulomatous inflammation, and fibrosis, which are essential for distinguishing AE from malignancies or other hepatic lesions. Additionally, it aids in assessing disease activity and guiding treatment decisions, especially in cases where imaging and serology yield inconclusive results. However, given its invasive nature and potential risks, the biopsy is typically reserved for diagnostically challenging cases.

(4) What is the importance of interdisciplinary diagnosis, for example in a multidisciplinary board?

Re: AE presents diagnostic challenges due to its tumor-like characteristics, requiring the integration of multiple disciplines for accurate assessment. Clinical laboratory experts perform serological and molecular tests to support diagnosis. Pathologists confirm the presence of parasitic structures, especially in cases mimicking malignancies. Radiologists contribute by identifying characteristic imaging features, while nuclear medicine specialists utilize PET-CT to distinguish active from inactive lesions. Additionally, surgical and interventional specialists assess resectability and alternative therapeutic options, while infectious disease specialists guide antiparasitic therapy. Public health professionals also play a vital role in surveillance and prevention strategies. By incorporating these perspectives, multidisciplinary team collaboration ensures comprehensive patient evaluation and facilitates personalized treatment plans.

(5) When is it a confirmed E. multilocularis infection?

Re: The diagnosis of alveolar echinococcosis requires a comprehensive approach, integrating clinical presentation, imaging, serological testing, and pathological examination. When a patient presents with relevant symptoms or imaging reveals a hepatic mass, alveolar echinococcosis should be suspected. Serological testing should be performed promptly following imaging, with positive results supporting the diagnosis. If imaging and serological tests cannot provide a definitive diagnosis, the pathological examination is considered the "gold standard". The identification of typical Echinococcus multilocularis structures, such as the germinal and laminated layers, on pathology, provides the final confirmation.

(6) Line 45: Echinococcosis in Austria is no longer restricted to the west, but is now widespread throughout the country. (Schneider R, Aspöck H, Auer H. Unexpected increase of alveolar echinococcosis, Austria, 2011. Emerg Infect Dis. 2013 Mar;19(3):475-7. doi: 10.3201/eid1903.120595)

Re: Thank you for pointing this out. We have changed "western Austria" to "Austria" in the main text and inserted the recommended reference in the corresponding position. Page 2, line 49. Reference 12.

(7) Line 64: Echinococcus multilocularis is not in italics.

Re: Thank you for your meticulous review. We have changed Echinococcus multilocularis in line 64 to an italicized format. Page 2, line 67.

(8) Line 83: In Figure 1 foxes and coyotes are listed as the most important final hosts, in the text foxes, canines such as dogs and raccoons (?), please standardise text and figure.

Re: We appreciate your suggestion. After reviewing the literature, we found that the main definitive hosts of Echinococcus multilocularis are foxes and, less commonly, dogs and raccoon dogs. To improve accuracy, we have added dogs and raccoon dogs as definitive hosts in Figure 1 and ensured consistency in the corresponding sections of the text. Page 2, line 52, Page 3, line 85−86.